4D street view: a video-based visualization method

Kageyama Akira kage@port.kobe-u.ac.jp
http://orcid.org/0000-0002-9210-467X Sakamoto Naohisa
Graduate School of System Informatics, Kobe University , Kobe , Japan
Mei Gang
Electronic publication date: 2020 Nov 9
Publication date: 2020
Volume: 6
Electronic Location ID: e305
Received 2020 Apr 29; Accepted 2020 Sep 29
Copyright: © 2020 Kageyama and Sakamoto
Copyright year: 2020
Copyright holder: Kageyama and Sakamoto
License: This is an open access article distributed under the terms of the Creative Commons Attribution License, which permits unrestricted use, distribution, reproduction and adaptation in any medium and for any purpose provided that it is properly attributed. For attribution, the original author(s), title, publication source (PeerJ Computer Science) and either DOI or URL of the article must be cited.
License URL: https://creativecommons.org/licenses/by/4.0/

Keywords: Scientific visualization, High performance computing, Computer simulation, In situ visualization, Video-based visualization, Omnidirectional visualization, Multi-viewpoint visualization, Interactive exploration of video dataset, New method for visualization of supercomputer simulations

Funding: Grant-in-Aid for Scientific Research (KAKENHI) 16K12434 and 17H02998 Support Center for Advanced Telecommunications Technology Research (SCAT) Foundation I-O Data Foundation Tateishi Science and Technology Foundation This work was supported by Grant-in-Aid for Scientific Research (KAKENHI) 16K12434 and 17H02998, SCAT (Support Center for Advanced Telecommunications Technology Research) Foundation, I-O Data Foundation, and Tateishi Science and Technology Foundation. The funders had no role in study design, data collection and analysis, decision to publish, or preparation of the manuscript.

==============================
We propose a new visualization method for massive supercomputer simulations. The key idea is to scatter multiple omnidirectional cameras to record the simulation via in situ visualization. After the simulations are complete, researchers can interactively explore the data collection of the recorded videos by navigating along a path in four-dimensional spacetime. We demonstrate the feasibility of this method by applying it to three different fluid and magnetohydrodynamics simulations using up to 1,000 omnidirectional cameras.

Introduction

Visualization is indispensable to comprehend the time development of complex phenomena captured as numerical data by supercomputer simulations. For three-dimensional (3D) fluid flow simulations, for example, pressure, velocity, vorticity, and other 3D fields are saved for post-hoc exploratory visualization. It is commonplace to save 3D fields without thinning out to maintain the spatial resolution of the simulation. In contrast, it is uncommon to store 3D fields without thinning them out in the temporal dimension, and the 3D data are usually output after very long intervals of time. If one attempts to save all the 3D fields within very short intervals, the numerical data will flood the disk space of the supercomputer system. The temporal thinning before the visualization often necessitates the discarding of valuable information. This situation will not improve in the future, but will most likely worsen as supercomputer technology advances.

We overcome the above-mentioned problems using our new in situ visualization method, wherein visual recordings are made using the supercomputer system during simulation. Because the recorded images are two-dimensional (2D), the data size is naturally smaller—a compression of the raw 3D data whereof the image is produced. In addition, lossless compression algorithms based on arithmetic coding are also available to reduce the size of the recordings even further.

In situ visualization already plays an essential role in large-scale simulations (Ma et al., 2007; Ross et al., 2008; Ma, 2009; Childs et al., 2012). Major application programs for general-purpose visualization are now armed with in situ libraries or tools, such as Catalyst (Ayachit et al., 2015) for ParaView and libsim (Whitlock, Favre & Meredith, 2011) for VisIt. These applications have been developed based on Visualization Tool Kit (VTK) (Schroeder, Martin & Lorensen, 2006). OpenGL is used for hardware-accelerated rendering in VTK when the graphics processing unit (GPU) is available. For a supercomputer system without GPUs, OpenGL functions can be executed using OSMesa. Software optimization technology, such as SIMD extension instructions for software rasterization processing, are included in OSMesa. Based on those technologies, a high-quality rendering framework, such as Embree (Wald et al., 2014) and OSPray (Wald et al., 2017), capable of executing rendering processing at high speeds have been developed. To execute the in situ visualization process efficiently on the supercomputing system, an adaptable data I/O framework called as ADIOS (Lofstead et al., 2008) has been developed. ADIOS provides a simple and flexible approach for data transmission, including asynchronous communication between the simulation and visualization processes (Wang et al., 2017; Kress et al., 2016). A generic in situ interface SENSEI (Ayachit et al., 2017) provides simulation researchers with portable interface for other in situ infrastructure such as Catalyst, libsim, and OSPray.

A shortcoming of conventional in situ visualizations is that they deprive users of interactive control. Here, interactive control means that users can, in real time, change visualization-related settings, such as the viewing position and direction. The interactivity also includes the capability to change the visualization method (e.g., from isosurface to volume rendering) applied to the target data and to tune the related parameters. In the conventional in situ visualization method, one has to resubmit the same simulation job to obtain a different visualization setting, for example, volume renderings from a different point of view.

There are several proposals to incorporate the interactivity into in situ visualization. Kawamura, Noda & Idomura (2016) developed an in situ visualization method that is an extension of the particle-based rendering method (Sakamoto et al., 2007) to the concurrent in situ visualization. In the particle-based rendering method, a dataset, represented as a set of particles, is produced by simulation. The particle dataset, whose size is much smaller than the raw numerical data, is transmitted to computer system for visualization with GPUs and interactively visualized. Tikhonova, Correa & Ma (2010a) and Tikhonova, Correa & Ma (2010b) proposed “proxy image” method to realize interactive volume rendering by making use of intermediate representation of the data. This method was extended to the in situ visualization by Ye, Miller & Ma (2013). Matthes et al. (2016) developed in situ library ISAAC for supercomputer simulations, by which users can interactively analyze data without the deep copy.

In our previous article (Kageyama & Yamada, 2014), we proposed an interactive in situ visualization method, in which we place visualization cameras on a spherical surface that covered the target region, as is schematically shown by the small circles in Fig. 1A. The simulation results are rendered with several visualization methods and parameters from the multiple viewpoints in the 3D space. Because the cameras focus on a point in the simulation region, as suggested by the arrows in the figure, we can observe the structures and dynamics in the simulation from various directions. The sequence of images are complied as videos that are indexed with the camera locations, before applying interactive analysis of the video collection with dedicated exploratory visualization software. This method can be regarded as an application of the image-based rendering (Shum & Kang, 2000) in computer graphics to the scientific visualization. In the image-based rendering, an image viewed from a specified point is calculated from images taken from different view points. Among various approaches to image-based rendering, our approach is closer to “light field rendering” (Levoy & Hanrahan, 1996), where new images are generated only through the image, not through the geometry data. Ahrens et al. (2014) also developed the image-based in situ visualization method and implemented as Cinema viewer, which is applied to various simulations (O’Leary et al., 2016; Banesh et al., 2017).

Figure 1 Visualization cameras (small white circles) and simulation boundary (rectangle).

(A) In our previous article (Kageyama & Yamada, 2014), we placed the cameras on a spherical surface around the simulation region. Arrows represent the viewing direction. All the cameras are focusing on a point in the simulation region. (B) In the present article, we propose to place omnidirectional cameras all over the simulation space. Each camera records the full solid view angle.

Because in situ visualization can produce a large number of images at fine time intervals, it is natural to save the sequence of images taken from a single camera as a single video file. A video file has an advantage over still images, that is, we can apply advanced video compression technology to it. The small storage requirement of a compressed video file allows us to save a large number of them (Berres et al., 2017), within the same disk space for storing raw numerical data. Therefore, we can set numerous visualization cameras with different visualization settings.

An increased number of video capturing virtual cameras implies an increased usage of the supercomputing resources, besides those necessary for the simulation. However, we note that it is not uncommon that only a portion of the available computing resources are utilized by a single simulation, even when the simulation code is highly optimized. In other words, the computing resource capacity of today’s supercomputer system is much larger than is being utilized by simulations. Thus, it would be quite appropriate to use surplus processor cores for our in situ visualization.

A natural extension of our previous approach (Kageyama & Yamada, 2014) is to place the visualization cameras all over the simulation region, that is, not only around it but also inside the region, as is schematically shown by the small circles in Fig. 1B. An apparent problem in this 3D distribution of the visualization cameras is that we cannot specify the view-direction of each camera when we have no a priori knowledge of the locations to be focused on.

To resolve this problem, we propose in this article to use omnidirectional visualization cameras that have the full (=4π steradian) solid view angle, as suggested by the arrows in Fig. 1B. If we could scatter such cameras all over the simulation region, almost all, if not all, occurring phenomena would be recorded by some nearby cameras.

Figure 2 schematically shows the concept of the proposed method. We place lots of omnidirectional cameras (blue spheres in Fig. 2A). Each camera produces a sequence of omnidirectional images, and we store them as a video file. When we place C cameras and apply V kinds of visualizations on each of them, we obtain C × V video files in total as a result of simulation (Fig. 2B).

Figure 2 Concept of 4D street view.

(A) We place lots of omnidirectional cameras (blue spheres) in the simulation space and apply in situ visualization with them. (B) Each camera records the time development of the simulation viewed from its position with 4π steradian and the omnidirectional images are compiled as a video. The video files are tied with the position data, and they constitute a video data collection. (C) A dedicated application program extracts a sequence of images from the video data collection as a smooth (regular, not omnidirectional) video on a PC window. The user interactively specifies the position, direction, and time (or frame) to extract images from the data collection.

After the simulation, it is possible to pick up one or more of the video files and play it on a PC with a commonly available video player. Although, the displayed images are distorted because they are captured by omnidirectional cameras. Supposing that the picked-up video is recorded by the c-th camera (0 ≤ c < C) for the v-th visualization method (0 ≤ v < V), herein referred to as movc,v, what the common video player does is to extract the sequence of images in movc,v and presents them on the PC’s screen one after another. In the “Virtual Museum” by Miller et al. (1992), the viewer interactively specifies the video to be played at one time. An important function in our method is that we extract a sequence of still images from different video files, such as movc1,v, movc2,v, movc3,v, ⋯, as denoted by the magenta arrows connecting Figs. 2A and 2B. The extracted sequence of the images is presented as a video by a dedicated application program (see Fig. 2C). The program excerpts a partial field-of-view around a user-specified direction from the omnidirectional image. The user’s experience is similar to walking around a specific region of interest.

We can also switch the currently applied visualization method, for instance, from va to vb, by exchanging the input video file from movc,va to movc,vb. This switching experience is almost the same as that of standard post-hoc visualizations. The difference, however, being that the exchange is quick in our method even if the switched visualization method vb is a computationally heavy one. The rendering has already been done on the supercomputer.

We can abstractly think of the video data collection as a set of videos with captured events from and within the simulation space. (We consider 3D simulations in this article). Since a video itself is a dataset of still images that are sequentially ordered in time, we can also regard the video data collection as a set of still images that are distributed in 4D spacetime. This data collection is a discrete subset of continuous function of pixel data defined in a five-dimensional space; the camera position in 3D and the viewing direction in 2D. Adelson & Bergen (1991) call the five-dimensional function “plenoptic function”, from plenus (complete)-optic.

In short, our method connotes sightseeing in 4D. Specifying a path in spacetime (see Fig. 3), we extract a sequence of images along the path. The images are presented on the PC window as a smooth video. If we notice an intriguing dynamics of the simulation in the video, we can fly to the location (i.e., we read the video file whose camera is close to the location) and observe the phenomenon. Note that we can also go back in time (or play the video backward in time) to identify the cause of the dynamics. We call this visualization method “Four Dimensional (4D) Street View” because it is reminiscent of Google Street View (Anguelov et al., 2010). Another ancestor of 4D Street View is QuickTime VR (Chen, 1995) that realized the interactive navigation of multiple files of panoramic pictures. The idea of interactive retrieval of image sequences from multiple video files can be traced back to “Movie-Maps” by Lippman (1980), in which the user interactively switch video segments at preregistered points.

Figure 3 Video data collections and a path in 4D spacetime.

Since the videos in the data collection are tied with their camera position, we can identify the data collection as a field of omnidirectional still images distributed in 4D (x, y, z and t) space. The user specifies a path in spacetime, and an application program extracts a sequence of images on the path and shows them on the PC window as a smooth video.

The actual procedure of the 4D Street View is separated into three stages, namely, the recording, conversion, and browsing. In the recording stage, we apply in situ visualizations on a supercomputer system with many omnidirectional cameras. In the conversion stage, we convert the output image dataset into a video data collection, which is the input of the browsing stage. In the browsing stage, we specify the camera path in 4D interactively and view the video on a PC window by a dedicated application program called 4D Street Viewer.

Recording Stage of 4D Street View

The recording stage of the 4D Street View should have the following three items, namely, multiple points of view, omnidirectional rendering, and in situ visualization.

Multi-point visualization

The multi-point visualization is theoretically the most straightforward part among the three items because it is generally possible to set plural viewpoints in an in situ visualization tool. In the 4D Street View, the configuration of the omnidirectional cameras is arbitrary. Herein, we place them in rectilinear distributions, that is, uniformly distributed in the x, y and z directions with the same spacings between the cameras in each direction. We denote the number of cameras in each direction as Cx, Cy and Cz. The total number of cameras is Cx × Cy × Cz.

Figure 4 shows two types of distribution of omnidirectional cameras (green spheres) that are used in test simulations described later. In (A) and (b), the camera configuration is (Cx, Cy, Cz) = (10, 10, 10), and in (c) and (d), it is (Cx, Cy, Cz) = (64, 1, 1). The rectangular boxes in the figure denote the boundary of the simulations.

Figure 4 Two samples of the omnidirectional camera configuration (green spheres).

(A & B) 3-D configuration of 1,000 cameras (viewed from different angles) used in the test simulation of the vortex-ring. (C & D) 1-D configuration of 64 cameras for the test simulation of Hall MHD turbulence.

Omnidirectional visualization

To apply omnidirectional visualization from each camera, we follow the standard procedure of the cube mapping (Greene, 1986; Sellers, Wright & Haemel, 2015). We assume a virtual sphere and a cube around the point; see Fig. 5A. Projecting the edges of the cube onto the sphere from the center, we have twelve arcs of great circles. The arcs divide the sphere or the 4π solid angle into six parts, that is, front, back, right, left, top, and bottom. We perform visualizations with the regular perspective projection six times for the six areas. As a result, we obtain six pictures and save them in the Portable Network Graphics (PNG) format using libpng library during the simulation. Note that PNG is a graphics format with lossless compression.

Figure 5 The omnidirectional visualization method in this study.

(A) We divide the solid angle 4π around a point of view into six parts by projecting the edges of a cube onto a sphere. The centers of the cube and sphere are on the point. We then apply the standard perspective projections for each part for six times. (B) Example of the six images obtained by the in situ visualization in a test simulation.

The pixel size of the six perspective projections used herein is 512 × 512. An example of the six images is shown in Fig. 5B. The six PNG files are combined to a single file as an omnidirectional image after the simulation in the conversion stage to be described later.

We can, of course, directly apply the conversion to the omnidirectional images during the simulation. Implementing such functions is one of the future topics of this research. It would also be possible to use a visualization tool with direct rendering with the omnidirectional projection.

In situ visualization

We can choose any in situ visualization strategy for the recording stage. Here, we take the synchronous approach wherein the visualization functions are called and controlled by the simulation program. The same approach is taken, for example, by Yu et al. (2010). In this synchronous in situ visualization method, the simulation and visualization programs share the computer nodes. The computation for the simulation is suspended while the visualization is running. There is another approach to in situ visualization, referred to as the asynchronous in situ visualization, as in Dreher & Raffin (2014). In this case, the visualization program runs independent of the simulation program. We will comment on this asynchronous in situ visualization in Conclusions.

As a renderer, we use VISMO (Ohno & Ohtani, 2014) in this study. VISMO is an in situ visualization library written in Fortran. It supports fundamental visualization methods, including the isosurface, slice plane, arrow glyphs, streamlines, and volume rendering. An essential feature of VISMO is that all the visualization methods are implemented based on the ray casting algorithm (Levoy & Marc, 1990). Therefore, VISMO can perform the in situ visualization on supercomputer systems with no GPU. VISMO is a self-contained library that requires no other basic tools or libraries except for libpng and a Fortran compiler. We can get visualization images with a simulation program by just calling VISMO’s functions. This is a great merit for simulation researchers. In our experiences, one of the general practical burdens in the situ visualization is the preparation of a necessary environment for a supercomputer system, for example, to install OSMesa that is a basic off-screen rendering library for OpenGL. Srend (Wetherbee et al., 2015) is a similar library as VISMO; it performs the in situ visualization of the volume rendering by the ray casting.

Data conversion to MP4

As described above, every in situ visualization from a fixed point of view produces a set of six PNG images. After the simulation, we combine the six files into a single image, according to the following procedure. Firstly, we assume that the six images are projected onto the sphere (see Fig. 4A). The spherical image is then mapped to a single rectangular image by the equirectangular projection. The pixel size for the omnidirectional image in this work is 2,048 × 1,536.

A set of omnidirectional image files from a certain camera position is then converted to a video file with MP4 codec (Puri, Chen & Luthra, 2004), which invokes a lossy but quality preserving compression to reduce the size. We use ffmpeg for this conversion.

Test Simulations

To test 4D Street View, we apply this method to three different kinds of simulations.

Thermal convection in spherical shell

The first test simulation is thermal convection in a spherical shell vessel (see Fig. 6). A fluid is confined between two concentric spheres of radii ro = 1.0 and ri = 0.835876. Inward central gravity (gravity acceleration g) and fixed temperature difference ΔT between the spheres drive the thermal convection motion. Convection in spherical shells have been studied extensively in geophysics and astrophysics. The test simulation in this article is characterized by the lack of rotation of the spheres and the relatively shallow depth of the shell compared to the standard geophysical and astrophysical simulations. It is known that thermal convection in a spheical shell with relatively small gap, ro − ri, exhibits very different patterns, such as spiral rolls (Zhang, Liao & Zhang, 2002; Itano et al., 2015), from that in a deep shell.

Figure 6 The simulation model of the spherical shell convection and the coordinate system.

We solve the Navier-Stokes equation with the finite difference method on the Yin-Yang grid (Kageyama & Sato, 2004). We denote the spherical shell region on the spherical coordinate system {r,ϑ,φ} as ri ≤ r ≤ ro, 0 ≤ ϑ ≤ π, −π ≤ φ < π. The grid size in the radial span is Nr = 21; the grid size in the latitudinal span of Yin- or Yang-component grid (π/4 ≤ ϑ ≤ 3π/4) is Nϑ = 240; the grid size in the longitudinal span (−3π/4 ≤ φ ≤ 3π/4) is Nφ = 720. The total grid size is Nr × Nϑ × Nφ × 2 = 21 × 240 × 720 × 2. (The last factor 2 is for Yin and Yang components.) A fourth-order explicit Runge-Kutta method is used for the time integration. Rayleigh number Ra = 2 × 104, which is a non-dimensional parameter that characterizes the drive of the convection.

The initial condition of the simulation is force-balanced, convectively unstable state, that is, ∇p0 = ρ0 g, where ρ0 and p0 are mass density and pressure at time t = 0. The initial flow is zero; v = 0 at t = 0. We put a perturbation δp to the initial pressure field, p = p0 + δ p(r, ϑ, φ), to start the convection. In this experiment, we set a single-mode of the spherical harmonics; δp(r,ϑ,φ)=a(r)Y~ℓm(ϑ,φ), where Y~ℓm is (normalized) spherical harmonics; ∫Y~ℓmY~ℓ′m′dS=δℓℓ′δmm′. The radial function is given by a(r)=c0sin⁡(π(r−ri)/(ro−ri)) with c0 = 5 × 10−2. We set l = m = 32 in this test simulation, which means that the perturbation δ p is highly localized in the (co)latitudinal direction ϑ, near the equator ϑ = π/2, while it has sinusoidal mode m = 32 in the longitudinal direction φ.

For 4D Street View, we place C = 9 omnidirectional cameras on the equatorial plane ϑ = π/2 with (Cx, Cy, Cz) = (3, 3, 1). We record the time development of the spherical shell convection with a constant interval of non-dimensional time τ = 0.173, from t = 0 to t = 14.5 (for every 100 simulation time-steps) for 85 frames in total. Figure 7 shows the time development of the convection visualized by isosurfaces of normalized radial velocity vr.

Figure 7 Thermal convection in the spherical shell visualized by isosurfaces of normalized radial velocity vr = −0.02: The banana-like purple objects indicate the downward flow in the convection.

(A)–(D) are convections at t = 0.346, 0.519, 2.08 and 6.58, respectively, viewed from a point on the equatorial ϑ = π/2 with radius r = 1.6, (E)–(H) are omnidirectional images at the same instances taken by the omnidirectional camera placed at the spherical origin r = 0.

Hall MHD turbulence

The second simulation for the test of 4D Street View is a magnetohydrodynamics (MHD) turbulence with the Hall term. We incorporate in situ, multi-point, omnidirectional visualization cameras into a Hall MHD turbulence simulation code (Miura & Hori, 2009; Miura, Araki & Hamba, 2016; Miura, 2019). In the simulation, the time development of the Hall MHD equations are solved by the Fourier pseudo-spectral method. The simulation geometry is a cube with the periodic boundary condition in all (x, y and z) directions with 256 grid points in each direction. The 2/3-truncation technique is used for the de-aliasing. The Runge–Kutta–Gill method is adopted for the temporal integration.

As the initial condition, large scale velocity and magnetic fields are specified with random phases. Just after the simulation starts, the fluid goes through instabilities and the velocity and magnetic energies are transferred toward smaller and smaller scales until it reaches to a fully developed turbulence. The highly complicated dynamics is a test bench of the true value of 4D Street View for the interactive analysis of the in situ visualization.

In this simulation, we try 1-D configuration of the omnidirectional cameras; (Cx, Cy, Cz) = (64, 1, 1) with 64 viewpoints on the x axis with a constant spacing, as shown in Figs. 4C and 4D. As shown in Fig. 8, we apply the in situ visualizations for the isosurfaces of the electric current density (colored in green-to-red) and the enstrophy density (gray-to-white).

Figure 8 The hall MHD simulations used as a test of 4D street view.

We apply three in situ visualizations (isosurface for two different fields and superposition of the two visualizatio methods). The upper three panels are for the initial condition. The time goes on to the middle and lower panels.

Vortex ring formation

The third test is simulation of vortex ring (or a smoke ring) formation, which is well know phenomenon; see for example, Shariff & Leonard (1992) and Lim & Nickels (1995) for reviews.

We assume a quiet gas in a rectangular box and apply an impulsive force in a localized region when the simulation starts. The fluid in the forced region is driven in the direction of the force. Although the initial flow exhibits complicated structures, the flow soon settles into a localized vortex in a simple torus, which is called a vortex ring. The vortex ring propagates at a uniform speed while maintaining its shape.

The simulation model is as follows: We solve the Navier-Stokes equation for in the cartesian coordinate system (x, y, z). The simulation region is a rectangular box with normalized side lengths of Lx × Ly × Lz = 20 × 10 × 10; see Fig. 9. The periodic boundary condition is assumed in all directions of x, y and z. The origin of the coordinate system is at the center of the simulation region. In the initial condition, the fluid has no flow with uniform temperature and density. We apply a pulse of force F to drive the fluid when the simulation starts at t = 0.

Figure 9 Simulation model of the vortex ring.

(A) shows the simulation model. We assume a rectangular region filled with a compressible fluid with normalized side lengths 20, 10 and 10 in x, y and z directions, respectively. The origin of the coordinate system is at the center of the region. A pulse force in the +x direction is applied in a cylindrical region near the end boundary at x = −10. The force drives the fluid in the cylinder and it soon settles into a ring-shaped vortex. (B–D) show vortex ring propagation in the +x direction.

The Navier-Stokes equation is discretized by a second order, central finite difference method. A fourth-order explicit Runge-Kutta method is used for the time integration. The periodic boundary condition is assumed for all (x, y and z) directions.

For this simulation, we have tried three different configurations of the omnidirectional cameras; (Cx, Cy, Cz) = (10, 10, 10), (20,10,5) and (3, 2, 1) with three visualization methods; (1) isosurface of the enstrophy density e = |∇ × v|2, where v is the flow velocity; (2) arrow glyphs of v; (3) stream lines of v.

We performed several experiments with different parameters for physics and visualization. In most cases, the driving force vector F has only the x component, but in one experiment, we also apply perpendicular (y and z) components in such a way that the driven flow has twisting component. It leads to an intriguing formation of the vortex ring with the flow helicity.

4D Street Viewer

The omnidirectional video files are tied with their camera positions by means of their file names. For example, a video file named sample.00123.mp4 or sample.03_02_01.mp4 is recorded by the omnidirectional camera located at (cx, cy, cz) = (3,2,1) in the rectilinear configuration (Cx, Cy, Cz). The video files termed according to this naming convention constitute the input data collection of the 4D Street Viewer.

The user specifies a camera position (cx, cy, cz) and changes it in real time through the 4D Street Viewer. Some keys on the keyboard are allocated to increment or decrement cx, cy and cz. Additionally, the user can specify the next camera to move by mouse clicks. A mouse click specifies a direction where one wants to move in the presented simulation space in the window. The 4D Street Viewer picks up one of neighboring cameras (cx ± 1, cy ± 1, cz ± 1) that is the closest to the line defined by the direction.

Each time the user specifies a new camera position or switches to a new visualization method, the 4D Street Viewer retrieves the corresponding file from the data collection. It imports one file at a time directly from the data collection without buffering or prefetching. Since the frame number in the video is consistently passed over to the next video, the user can smoothly change the video file while playing a video.

Figure 10 shows a snapshot of the window of the 4D Street Viewer with its user interface. One can resize the window by dragging a corner of the window with the mouse. Shown in the left panel is the view of the simulation from the current camera position and direction. In the right panel of the window, some texts are placed to present input video and the current status. The playing mode of the video (play/stop/reverse) as well as the current frame are controlled by the buttons and slider bars in the right panel.

Figure 10 A snapshot of the window of the 4D street viewer.

The left panel shows the current view extracted from an omnidirectional image in the input video. The camera configuration (Cx, Cy, Cz), current camera position (cx, cy, cz), current number of frame, and other information of the video data collection are shown in the right panel.

We have developed a 4D Street Viewer in the framework of KVS (Sakamoto & Koyamada, 2015). KVS is a visualization development framework provided as a C++ class library. Although the primary purpose of KVS is to provide a modular programming environment for scientific and information visualizations, it can also be used as a more general framework for the development of such a 4D Street Viewer as this.

Interactive Analysis of the Test Simulations with 4D Street View

Viewpoint translation

We first present an interactive translation of the viewpoint by 4D Street Viewer, applied to the Hall MHD simulation. Figure 11 is a sequence of six snapshots from a recorded video in Supplemental Data (https://github.com/vizlab-kobe/4DSVTestData/blob/master/movies/03-hall-mhd.mov). Since the number of the omnidirectional cameras along the axis is sufficiently high (64 cameras), the user has a smooth experience of walk-through by changing the viewpoint by means of slide bar, mouse click or keyboard input. Although the six images (A)–(F) show the same frame (t = 0) in this Fig. 11, the video in Supplemental Data present the interactive translation also in time in the simulation’s space-time.

Figure 11 Translation of the viewpoint in the Hall MHD simulation.

(A–F) show an image sequence in the walk-through in the simulation region.

Note that the smooth experience of the walk-through by the viewpoint translation by 4D Street View is not degraded even if the visualized scene is much more complicated when the turbulence is fully developed. (See the later part of the Supplemental video.) On the other hand, if we applied the post-hoc visualization for the fully developed turbulence, such as the bottom three panels in Fig. 8, the response would be slowed down for there are myriads of polygons.

Looking around

As in standard visualization applications, a mouse drag is allocated to the rotation of the view in the 4D Street Viewer. In our case, a rotation means resampling of the field-of-view around the specified direction from the omnidirectional image, as in the regular video-playing programs for panoramic videos. We perform the resampling with a GPU.

In the standard post-hoc visualization programs, a rotation specified by a user invokes a rotational transformation of the scene in the computer graphics. Getting a new image after a rotation may be time-consuming if the rotated scene requires a massive rendering. As in the case of the viewpoint translation, in the 4D Street View, a rotated image always appears momentarily, regardless of the complexity of the scene, since views from any angle are already stored in the omnidirectional video.

Figure 12 is a snapshot sequence taken from a video in Supplemental Data (https://github.com/vizlab-kobe/4DSVTestData/blob/master/movies/04-thin-shell-convection.mov). This figure demonstrates the changing the viewing-direction by 4D Street Viewer, for the spherical shell convection simulation. In the the left three panels, the viewpoint is located at a point on the equator at r = 0.807, which is just inside the inner sphere r = ri. The purple bar-like objects are isosurfaces of negative velocity vr; they denotes the downward flow of the convection cells (or convection rolls). The time frame is fixed in the left three panels.

Figure 12 Change of viewing direction of the spherical shell convection in the 4D Street Viewer by the mouse drag motion.

In the right three panels of Fig. 12, we move to the viewpoint located at the origin r = 0. We then play the video forward in time for a while to observe that the bar-like convection rolls grows toward the north and south poles, applying the mouse-drag again to observe the ring-like patterns near the north pole. The bar-like structure of the convection rolls cannot be maintained in the high latitude regions because the distances between the neighboring rolls (in the longitudinal direction) are decreased as the tips of the rolls get closer to the poles. Instead of the longitudinal rolls in the low-latitude regions, ring-like pattern appears in the polar regions, which is observed by isosurface of positive vr (upward flow) in the right three panels.

Change of Visualization Method

We have changed the visualization method in 4D Street Viewer from the left panels to the right panels in Fig. 12. Figure 13 shows another example of the interactive switching of the applied visualization methods. This figure is an image sequence taken from a video in Supplemental Data (https://github.com/vizlab-kobe/4DSVTestData/blob/master/movies/02-vortex-ring.mov). In this case, simulation of the vortex ring with the twisting force is visualized by (i) isosurface of enstrophy density e and (i) isosurface of e plus stream lines of the flow velocity v. In this visualization, we change time frame, viewpoint position, viewing direction, and visualization method (with and without the stream lines).

Figure 13 The vortex ring simulation with the twist force.

The vortex ring simulation with the twist force. Two visualization methods (with and without the stream lines) are switched while playing the video. (A) shows an isosurface visualization without the stream lines. (B–E) show the same time frame with different view position and direction. (E) and (F) are also from the same frame later in the time development.

Discussion

The MP4 conversion by ffmpeg from the PNG images is the only stage wherein lossy compression is applied in our procedure. (The mapping to an omnidirectional image from the six directional images described incorporates pixel interpolations, but the image deterioration by the interpolation is negligibly small). To confirm the impacts of the lossy compression by MP4, we compare, in Fig. 14, an omnidirectional PNG image and an image extracted from the MP4 video by 4D Street Viewer at the same frame. The degradation of the image quality by the MP4 conversion is negligibly small for visualization analysis.

Figure 14 Comparison of images before (A) and after (B) the lossy compression by MP4 codec.

(A) The original omnidirectional PNG image. (B) Snapshot from MP4 video file at the same frame.

In this work, the largest number of the omnidirectional cameras scattered in the test simulations was 1000, that are for the vortex ring simulation. We applied two visualizations from each camera and the total size of the video files (2,048× 1,536 pixels × 101 frames) compressed in the MP4 codec was 6.0 × 109 B.

This result shows that the data size of the video data collection in the 4D Street View with a thousand cameras can be regarded as small in today’s capacity norm of storage devices. The small size of the video data collection is due to the compression by the MP4 codec.

For the Hall MHD simulation, we performed with the spatial resolution of 2563 and applied the in situ visualization for 21 time frames. The total storage size to save the three kinds of omnidirectional MP4 video files (electric current, enstrophy density, and their superposition) are 1.8 × 109 B. On the other hand, if we are to perform the post-hoc visualization, we would have to save raw numerical data of at least four fields (three components of the velocity vector plus one scalar) for the double precision (8 B) simulation. The storage size amounts to 2563 × 4 (fields) × 21 (time steps) × 8 B ∼ 1.1 × 1010 B.

Even in this small-scale simulation, the storage size for 4D Street View method is smaller than the post-hoc visualization. The spatial resolution 2563 is too low in the current standard of the high performance computing. It is not rare to perform turbulence simulations with 4,0963 resolution these days. Saving the raw numerical data for the post-hoc visualization with this spatial resolution (and with corresponding temporal resolution) is certainly impractical: 4,0963 × 4 (fields) × 336 (time steps) × 8 B ∼ 7.4 × 1014 B. On the other hand, the storage size for 4D Street View is only weakly dependent on the spatial resolution of the simulation.

In this work, we have placed the omnidirectional cameras in rectilinear configuration of 1-D, 2-D, or 3-D. However, the camera density is not necessarily uniform in the 4D Street View. We can reduce the total number of cameras by distributing them in a nonuniform way, concentrating only near focused regions.

The video collection explored in the 4D Street View is a set of discrete samples of continuous plenoptic function (Adelson & Bergen, 1991) in the 5D space. It would be possible to apply image-based rendering techniques in computer graphics such as “view interpolation” (Chen & Williams, 1993) or “Plenoptic Modeling” (McMillan & Bishop, 1995) to obtain visualized images viewed from intermediate positions between specified view points.

Generally, in situ visualization is a way to use supercomputer resources to suppress the burden of simulation researchers. They have recently spent most of their research time on post-hoc visualization, such as data transfer and preparation before starting the visualization itself. The situation will become worse in the future with the further development of supercomputer technology. Even today, the computing power for a single supercomputer system is excessive for a single simulation job. It would be a valid idea to use a supercomputer system primarily for the visualization and secondly for the simulation.

Conclusions

We have proposed a new visualization method referred to as the “4D Street View” for large-scale simulations. The key idea is to record a simulation with many omnidirectional cameras that are scattered in the simulation space. The cameras record the simulation through in situ visualization. As a result of the simulation, we obtain a data collection of the omnidirectional videos. The videos in the data collection are tied with their camera positions. The video data collection can be regarded as a field of omnidirectional still images in 4D spacetime.

We have developed a dedicated application program (4D Street Viewer) to explore video data collection. With the program, we interactively specify a path in 4D spacetime and extract a sequence of images along the path and show them as a smooth video on the screen.

If we find an intriguing phenomenon that is far away from the current camera, we can “fly” there with the 4D Street Viewer and scrutinize it from nearby cameras. It is also possible to go backward in time to investigate the cause of the phenomenon. We can change the view-angle anytime since all views are stored in the omnidirectional images. We can also switch the applied visualization methods as long as the corresponding videos have been recorded in the simulation.

One of the challenges of the 4D Street View is the time spent on the visualization. We have applied the synchronous in situ visualization in this work; the same computer nodes are allocated for the simulation and the visualization. The parallelization of the simulation determined the number of nodes. When we perform a visualization-intensive computation, we should allocate much more nodes to the in situ visualization than the simulation. We are developing another approach—asynchronous in situ visualization—for the recording stage of the 4D Street View (Kageyama, Sakamoto & Yamamoto, 2018). The simulation program and the visualization program run independently on different computer nodes with no explicit synchronization. They have their own main programs and own MPI_Init and MPI_Finalize calls. To realize this multiple program, multiple data (MPPD) framework, we place a layer called Membrane between the simulation and the visualization. With this the Membrane method, we can allocate any number of computer nodes for the in situ visualization of visualization-intensive cases.

We would like to thank Prof. Chandrajit Bajaj for suggesting the use of omnidirectional cameras to improve our mutil-point in situ visualization method. We thank Ms. Keiko Otsuji for her skilled technical assistance. We are grateful to Prof. Nobuaki Ohno for providing us with the VISMO’s source code. We are also grateful to Prof. Hedeaki Miura for providing us with the Hall MHD simulation code.

Additional Information and Declarations

Competing Interests

Author Contributions

Data Availability

The authors declare that they have no competing interests.

Akira Kageyama conceived and designed the experiments, performed the experiments, analyzed the data, performed the computation work, prepared figures and/or tables, authored or reviewed drafts of the paper, and approved the final draft.

Naohisa Sakamoto analyzed the data, performed the computation work, authored or reviewed drafts of the paper, and approved the final draft.

The following information was supplied regarding data availability:

Source code is available at GitHub:

https://github.com/vizlab-kobe/4DStreetViewMovieViewer.

Data are available at GitHub:

https://github.com/vizlab-kobe/4DSVTestData.

Sample movies are available at GitHub:

https://github.com/vizlab-kobe/4DSVTestData/tree/master/movies.

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
