# Peer review of "D street view: a video-based visualization method"

_PeerJ Computer Science, doi:10.7717/peerj-cs.305_

## Round 0.1 · original submission · Major Revisions

Dear authors, Thank you for submitting to PeerJ Computer Science.

Overall, the paper is good. Please specifically focus on the following issues when revising your manuscript.

(1) please add the video of operating the 4D street view application, or measured latency time for changing the views.

(2) The sample problem of Voltex simulation seems to be not suitable to demonstrate the impact of the 4D street view, because the iso-surface and glyphs in Fig. 10 to 13 are too simple. Please improve it.

(3) Please clearly indicate the storage space needed for a set of precomputed videos compared to the space needed for raw 3D visualization data.

(4) The authors mentioned the use of compression of video data in the set. But this approach leads to the loss of information since small changes may be lost after the compression is applied. Please can you discuss this?

Reviewer 1 ·

Basic reporting

This paper proposed a new visualization scheme named “4D Street View". In general, this paper is well organized and explains the new method comprehensively.

Experimental design

The analysis of the huge simulation data is a well-known problem. I agree that the new technology presented in this paper contributes to improving this difficulty. The concept of the 4D street view, which enables walkthrough inside huge omnidirectional movie files, is new. The paper presents the details of their implementation sufficiently.

Validity of the findings

The following points potentially fail to meet the requirement of this journal.

(i) Frequently, the author argues the smoothness of changing videos or operations as the advantages of their video management without showing objective data. Thus, it sounds subjective to me, a bit. To improve the presentation I suggest adding the video of operating the 4D street view application, or measured latency time for changing the views.

(ii) The sample problem of voltex simulation seems to be not suitable to demonstrate the impact of 4D street view, because the iso-surface and glyphs in Fig. 10 to 13 are too simple. The postprocessing visualization with such as Paraview and geometry files may utilize such visualization more flexibly.

(iii) As for (ii), the author may claim the large data size of raw data in line 414 in comparison to the data size of videos for 4D street view. However, some recent community software support to output not only the raw data but also the geometry (polygon) data. Then we can analyze the output geometry files in the post process visualization in a more flexible way than 4D street view. Thus, for fairness, the data size of movies and polygon data should be compared to justify the proposed scheme. Since the data size of the geometry increases with the complexity of objects, it was difficult to see why 4D street view is useful with the simple geometry objects in the demonstration.

(iv) Data size comparison around line 414 should be involved in the discussion or result section, not in conclusion.

·

Basic reporting

The language of the paper is clear and easy to understand for a specialist. I have found a small mistakes in the text, like a missing numbers in the references to Sections. so I recommend authors to check the text before publication.
The graphical data is clear and of good quality.

Experimental design

The research is within the scope of the journal. The main topic is in the field of scientific visualization and computer graphics.

The paper does not describe experimental research, so nothing can be said about the experiment design. The example chosen by authors to show the performance of the method is simple, but it shows the results.

Validity of the findings

The proposed solution is relevant to any fields of science and engineering where there is a need for 3D simulations of physical fields. It may be easily applicable to other areas of human activity - virtual tourism, education etc.

Additional comments

The paper presents a method of scientific simulation data visualization. Authors developed a method for visualisation of the 3D simulation results, which may find an application in the fields like fluid mechanics, atmospheric and ocean science, or any other were there is a need to visualize a complex 3D fields of physical quantities. One of the main contributions of the proposed method is an application of omnidirectional cameras for visualization which replaced the standard projective model. Another important impact is a development of a tool for post-visualization. In the new approach, when a user wishes to change the viewing angle while looking at the graphics, the new view is not recomputed, but downloaded as a frame from a precomputed set of video data. This approach will surely speed up the process of an analysis. However, there are issues which should be addressed.
1) Clearly the approach would benefit any post-simulation analysis of results. But a question is what is the storage space needed for a set of precomputed videos compared to the space needed for raw 3D visualization data? Its important in the case of high-resolution video and small steps of simulation.
2) Authors mentioned a use of compression of video data in the set. But this approach leads to the lose of information since small changes may be lost after the compression is applied. Can authors comment on this?
3) If the video set is predefined at the time of the simulation how one can smoothly change the views while looking at results? There will be jumps visible to a user. For simpler scenes it won't be an issue if one uses traditional real-time computer graphics. Can you considered some kind of blending of views when the number of cameras per volume is not enough to provide smooth transition between views?

---

## Round 0.2 · accepted · Accept

I am happy that the manuscript is well revised! Congratulations!

Reviewer 1 ·

Basic reporting

The revised manuscript properly responded to reviewer comments.

Experimental design

no comment

Validity of the findings

no comment

·

Basic reporting

The revised version of the manuscript contains answers to the reviewer's questions asked in the previous review report. Therefore, I think that the paper is suitable to be published in the journal. Also, the overall structure of the article has been improved. New simulations and provided graphics give an additional insight in the presented topic.

Experimental design

The new version of the manuscript shows an importance of the proposed method for the scientific visualisation of the simulation results. Three topics has been covered: thermal convection in spherical shell, magnetohydrodynamics turbulence phenomena and vortex ring formation. Topics are well chosen to present the benefits of the new visualisation method - the included graphics shows .

Validity of the findings

I think the choice of magnetohydrodynamics simulation for an assessment of the proposed visualisation method is correct. The included graphics shows the complexity of the phenomena which can be better understood is one has a possibility of the view manipulation of the proposed visualisation tool.

Additional comments

The revised version of the manuscript contains answers to the reviewer's questions asked in the previous review report. Therefore, I think that the paper is suitable to be published in the journal. Also, the overall structure of the article has been improved. Provided graphics with results of simulations gives additional insight in the presented topic.